# Forecasting ocean acidification impacts on kelp forest ecosystems

**Adam J. Schlenger**[1]*, **Rodrigo Beas-Luna**[2], **Richard F. Ambrose**[1,3]

1 Institute of the Environment and Sustainability, University of California Los Angeles, Los Angeles, California, United States of America, 2 Facultad de Ciencias Marinas, Universidad Autónoma de Baja California, Ensenada B.C. Mexico, 3 Department of Environmental Health Sciences, University of California, Los Angeles, Los Angeles, California, United States of America

☉ These authors contributed equally to this work.

* aschleng@ucla.edu

**Data Availability Statement:** All of the data can be found in an open access repository on GitHub using the following URL: https://github.com/rbeas/isla_natividad_ewe.

## Abstract

Ocean acidification is one the biggest threats to marine ecosystems worldwide, but its ecosystem wide responses are still poorly understood. This study integrates field and experimental data into a mass balance food web model of a temperate coastal ecosystem to determine the impacts of specific OA forcing mechanisms as well as how they interact with one another. Specifically, we forced a food web model of a kelp forest ecosystem near its southern distribution limit in the California large marine ecosystem to a 0.5 pH drop over the course of 50 years. This study utilizes a modeling approach to determine the impacts of specific OA forcing mechanisms as well as how they interact. Isolating OA impacts on growth (Production), mortality (Other Mortality), and predation interactions (Vulnerability) or combining all three mechanisms together leads to a variety of ecosystem responses, with some taxa increasing in abundance and other decreasing. Results suggest that carbonate mineralizing groups such as coralline algae, abalone, snails, and lobsters display the largest decreases in biomass while macroalgae, urchins, and some larger fish species display the largest increases. Low trophic level groups such as giant kelp and brown algae increase in biomass by 16% and 71%, respectively. Due to the diverse way in which OA stress manifests at both individual and population levels, ecosystem-level effects can vary and display nonlinear patterns. Combined OA forcing leads to initial increases in ecosystem and commercial biomasses followed by a decrease in commercial biomass below initial values over time, while ecosystem biomass remains high. Both biodiversity and average trophic level decrease over time. These projections indicate that the kelp forest community would maintain high productivity with a 0.5 drop in pH, but with a substantially different community structure characterized by lower biodiversity and relatively greater dominance by lower trophic level organisms.

**Funding:** RA Award number is C0100400 Ocean Protection Council https://www.opc.ca.gov/ The funders had no role in study design, data collection and analysis, decision to publish, or preparation of the manuscript.

**Competing interests:** The authors have declared that no competing interests exist.

## Introduction

Assessing the impact of climate change on marine ecosystems is a challenging task due to the complexity of interacting variables involved as well as the large spatial and temporal scales that these changes take place over. Natural equilibria are being affected by a wide range of environmental and anthropogenic pressures including temperature, deoxygenation, eutrophication, fisheries, and ocean acidification [1]. These chemical and physical changes influence marine communities on a variety of scales from organismal physiology to population change and ecosystem-scale structure and function [2]. Approaching a comprehensive understanding of these dynamics is made more difficult by the formidable requirements of measuring ecosystem change across all these dimensions simultaneously [3]. Despite such hurdles, significant progress is being made across multiple disciplines and those efforts are being combined to advance our understanding of ecological change in the face of global warming and anthropogenic development [4].

Studying the influence of environmental stressors on marine ecosystems is conducted through a variety of methods that have distinct tradeoffs with respect to forecasting ecosystem-scale responses to future climate change. One commonly used approach is to analyze paleo-records of marine organisms to assess the impact of changing climate conditions, which may serve as a proxy for predicting ecosystem responses under global warming [5, 6]. A major drawback of this approach is the fact that ecosystems have evolved over long periods since the formation of these records and their response to environmental change could differ, but also because there are a variety of other variables influencing these records that are not preserved with the same degree of accuracy (e.g. nutrient availability). Laboratory studies serve as an optimal way to isolate the effects of specific environmental stressors, or their interactions, and identify the physiological mechanisms through which they impact organisms [7, 8]. This approach is well suited for understanding how environmental stressors like temperature, oxygen, and ocean acidification influence metabolism, reproduction, and survival, but lacks the realism associated with trophic interaction effects that take place in natural ecosystems. Mesocosm and field studies can be used to fill this gap [9, 10], which include a more encompassing understanding of how community dynamics shift with environmental stressors. However, mesocosm studies generally do not include the full set of species and lack the spatial scales necessary to recreate the nonlinear dynamics of ecosystem-level change, and field studies have very little control over stressor gradients. Finally, ecological monitoring programs have proven to be a successful tool to enhance our understanding of climate change effects on natural ecosystems, contributing disproportionally to policy and management than any of the previous examples [11, 12]. However, these are demanding, costly, and it is difficult to identify the causation underlying patterns as one can generally only measure correlations.

Ecological models have the capacity to integrate results from all of these different methods. In addition, ecological modeling is well suited for making quantitative predictions of marine ecosystem change under future climate conditions [13]. These models can be used to evaluate the system-level consequences of environmental change by scaling up physiological relationships to populations and entire ecosystems. Data from field and laboratory studies can be incorporated into these models and combined with food web dynamics to create a mechanistic understanding of how ecosystems respond to external forcing. A major advantage of this approach is the ability to identify indirect, cumulative, and second order effects that are often missed or unable to be measured in field or laboratory settings [14]. The importance of indirect effects in driving ecosystem dynamics has been linked to a wide variety of mechanisms including the evolution of selective traits across species [15], coupled ecosystem interactions and energy flow [16], and community stability in the face of climate perturbations [17]. Given

the far-reaching consequences of global climate change, assessing species and population vulnerabilities requires a quantitative understanding of indirect effects [18], which serves as the foundation for effective mitigation strategies [19].

'End-to-End' ecosystem models that tie together climate, physical, and chemical aspects of an ecosystem to biotic and anthropogenic interactions are an increasingly important and widely used tool in quantifying the indirect ecosystem effects of climate change [20, 21]. Yet while the field of ecosystem modeling computation continues to advance, the extensive data collection needed to build and drive them is lagging. Considering end-to-end models encompass a broad scope of physical, chemical, and biological components, expansive and long-term field monitoring datasets are necessary for proper validation and accuracy [22, 23]. Furthermore, detailed physiological responses to environmental conditions across functional groups and trophic levels are needed to optimally parameterize ecosystem responses. Due to the difficulty of creating such comprehensive and multidisciplinary datasets, the ability to fully model and predict the interacting effects of multiple environmental stressors under future climate change is still a major challenge [24]. Therefore, using ecosystem models to forecast the effects of climate change can be made more effective by focusing on individual stressors for ecosystems that have extensive data available. Fortunately, ecosystem modeling can simultaneously be used to identify gaps in information to guide monitoring programs on how to best structure data collection to be more efficient, informative, and potentially cheaper, which will help make parameterization of interacting environmental constraints more readily available.

Ocean acidification (OA) is a global issue that is quickly becoming a major threat to ecosystem health across the world's oceans [25]. As carbon dioxide concentrations increase due to anthropogenic emissions, critical chemical equilibria in ocean surface waters are being disrupted, leading to consistent decreases in pH and inhibiting the ability for calcifying organisms to grow, survive, and reproduce [26]. These physiological effects are already beginning to manifest at the community level, impacting ecosystem health and industries that depend on ecosystem services [27]. Although many of the expected effects of OA are negative, direct effects of OA vary and for some taxa, such as non-calcareous seaweeds, are expected to be generally positive [28, 29]. Predicting ecosystem responses to increasing OA in the face of climate change has become a primary international concern for scientists and managers. Despite this importance, the study of OA is still a relatively new field and very little is known regarding its impacts on ecosystem properties, structure, function and the services provided.

While various paleo records [30], laboratory experiments [31], and field studies [32] have been used to predict the impacts of OA on specific species or functional groups, very few studies have investigated the mechanisms through which OA influences ecosystem dynamics [33]. Modeling efforts are being used to fill this gap by quantifying organism OA response curves [34] to parameterize ecosystem models. This has allowed researchers to identify ecosystem vulnerabilities in areas like the California Current [35], predict ecological change in the north Atlantic [36], and assess optimal management solutions [37]. However, a major drawback of these modeling efforts is the inclusion of only a single mechanism through which OA impacts model components, such as species production or mortality. OA affects a broad range of biological functions including metabolism, predator-prey interactions, and reproduction, and modeling studies have yet to quantify their cumulative effects.

Furthermore, these efforts are made more difficult due to the scarcity of comprehensive ecosystem-scale datasets and therefore encompass only a select group of ecosystems, generally in open ocean environments. Yet nearshore coastal ecosystems serve as ideal study sites for the effects of OA due to their exposure to the physical and chemical variability of both surface and deeper waters, the complex ecosystems that form there, and the economic value they provide [38]. Their proximity to human development and easy access also make them common

locations for scientific research, leading to robust and long-term datasets for many areas. Due to the importance of these ecosystems, their vulnerability to OA, and the existence of extensive field data for marine communities, nearshore coastal ecosystems provide an opportunity to model the ecosystem-level effects of climate change.

This study utilizes a long-term dataset and detailed food web model of a nearshore kelp community to quantify the impact of OA on marine ecosystem structure and function. This work builds upon past research by quantifying the individual and combined influence of multiple OA forcing mechanisms on population abundances as well as how those changes scale to impact ecosystem health, complexity, and the economic value of ecosystem services. This is accomplished by 1) identifying the differences between OA forcing of production, mortality, and predation, 2) simulating the impact of OA across various species and functional groups of a coastal kelp community, and 3) measuring changes in emergent properties related to ecosystem health, complexity, and economic output. We have chosen to model a kelp forest community because of the economic and ecological importance of kelp forests [39]. This research represents a quantitative modeling approach to assess the cumulative effects of multiple OA impacts on ecosystem dynamics. A major goal of this work is to provide a general understanding of macro-ecological responses and to highlight opportunities to strengthen field and laboratory validation of interacting effects of OA on marine communities.

## Methods

### Model development

The ecosystem model used in this study was developed using Ecopath with Ecosim (EwE) [40, 41]. EwE is a software package that comprises several modules (Ecopath, Ecosim, and Ecospace) for building mass balance as well as time and space dynamic models of ecosystems and includes a large set of diagnostics for analyzing food webs. These features, plus its flexibility and user-friendly interface, make EwE one of the most widely applied modeling approaches for food web representation and scientific analysis [42, 43]. The Ecopath component provides a platform for building static ecological networks (that is, food webs) where network components are defined as functional groups (representing species or groups of species) and the interactions between components are trophic interactions quantified using bioenergetics [44]. The major inputs for each component of the network include biomass, production and consumption rates, food preferences, unassimilated fraction, and fishing catches. Additional inputs include immigration, emigration, and biomass accumulation, while mortality and respiration are estimated within the model. Values for these parameters typically represent yearly averages over a small number of years.

Ecopath calculates a static mass-balanced snapshot of the biomass and energy fluxes between functional groups in a food web. The mass balance is set in a way that consumption of any given component is subtracted by all losses including bioenergetic ones (respiration, excretion), predation, eventual net migration, and mortality due to anthropogenic factors such as fishing. Remaining biomass can accumulate or result in an estimate of other mortalities unexplained by the model (summarized into the parameter Ecotrophic Efficiency; [44]). Ecosim extends the Ecopath model by providing a dynamic simulation capability at the ecosystem level using a system of differential equations that express biomass flux rates between pools as a function of time. Ecosim can be used to simulate ecosystem behavior through the incorporation of physical, biological, or anthropogenic forcing parameters.

We developed an EwE model for this study due to the practical considerations associated with developing, running, and analyzing a suite of OA simulations as well as the availability of previous OA studies using this framework for comparison [36, 45, 46], providing a foundation

to build upon and compare results with. Only a few ecosystem models have been published for kelp forest ecosystems [46–51]. While other quantitatively robust ecosystem modeling software packages are available for this type of ecosystem assessment, such as Atlantis [52], the Atlantis model for the California Current does not include a nearshore kelp community component and developing one would have been a multi-year effort falling outside the scope of this project.

## Ecopath model

Two temperate coastal ecosystem models developed for the Pacific coast were available for use; one for Monterey Bay, California, United States [47] and the other for Isla Natividad, in Baja California Mexico, near the southern distributional limit of giant kelp (*Macrosystis pyrifera*) [51]. The Isla Natividad model was selected for this study due to the longer history of development and associated publications as well as for the existence of a comprehensive ecological monitoring program dataset to support it. Isla Natividad is located in the Mexican Pacific in the middle of the Baja California Peninsula (Fig 1) [51]. It is a 7 km long island and is part of the Natural Protected Area of El Vizcaino. The model area covers 31.42 km$^2$ of temperate reefs, stretches from the coastline (0m) to the 30m isobath, and is predominately made up of kelp forest. The island is inhabited by approximately 300 people, who have been subsisting on marine products for more than 75 years [53]. Until 2010, the principal harvested species were the pink and green abalone (*Haliotis corrugata* and *H. fulgens*, respectively). These abalone species have recently suffered a drastic decline in their populations compared to historical records [54].

The Isla Natividad Ecopath food web is comprised of 40 functional groups including birds, marine mammals, fish, invertebrates, algae, phytoplankton, and zooplankton (Fig 2). Data used to parameterize this model was provided through an ecological monitoring program that began in 2006 through the non-governmental environmental organization Comunidad y Biodiversidad A.C. (COBI), the fishing cooperatives, and researchers from Stanford University. The model consists of four major trophic levels with *Macrocystis pyrifera*, *Ecklonia arborea*, snails, urchins, and wrasses, such as the California sheephead, having important ecological roles (Table 1). Due to the strong coupling of urchins with giant kelp and their influence on ecosystem stability [55], three urchin functional groups were included in the model (purple, red, and black urchins). Black urchins include both *Centrostephanus coronatus* and *Arbacia stellate*. Giant kelp and snails have the largest biomass of the species and groups included in the model. Basses, elasmobranchs, sheephead, marine mammals, sea birds, black urchins, and lingcod are keystone groups (i.e., high impact per unit biomass).

A summary of statistics related to ecosystem structure and function can be found in Table 2. In general, the summary statistics for Isla Natividad model fall within the average range of published Ecopath models [56]. The values of major energy flows per km$^2$ (e.g., production, respiration, total system throughput) are comparable with existing models of coastal shelf, open water, estuarine, upwelling, and island ecosystems. However, biomass per km$^2$ for this system is on the higher end. This is to be expected for nearshore coastal ecosystems, which tend to have a higher density of biomass compared to more open water systems, which make up the majority of published Ecopath models. The Isla Natividad model also has lower connective index and omnivory index values than most models, meaning that this ecosystem displays a less 'web-like' structure with lower levels of connectivity between species and tighter trophic interactions. This structure is also more common of nearshore coastal communities in comparison to open ocean, upwelling, or estuarine ecosystems.

## Isla Natividad

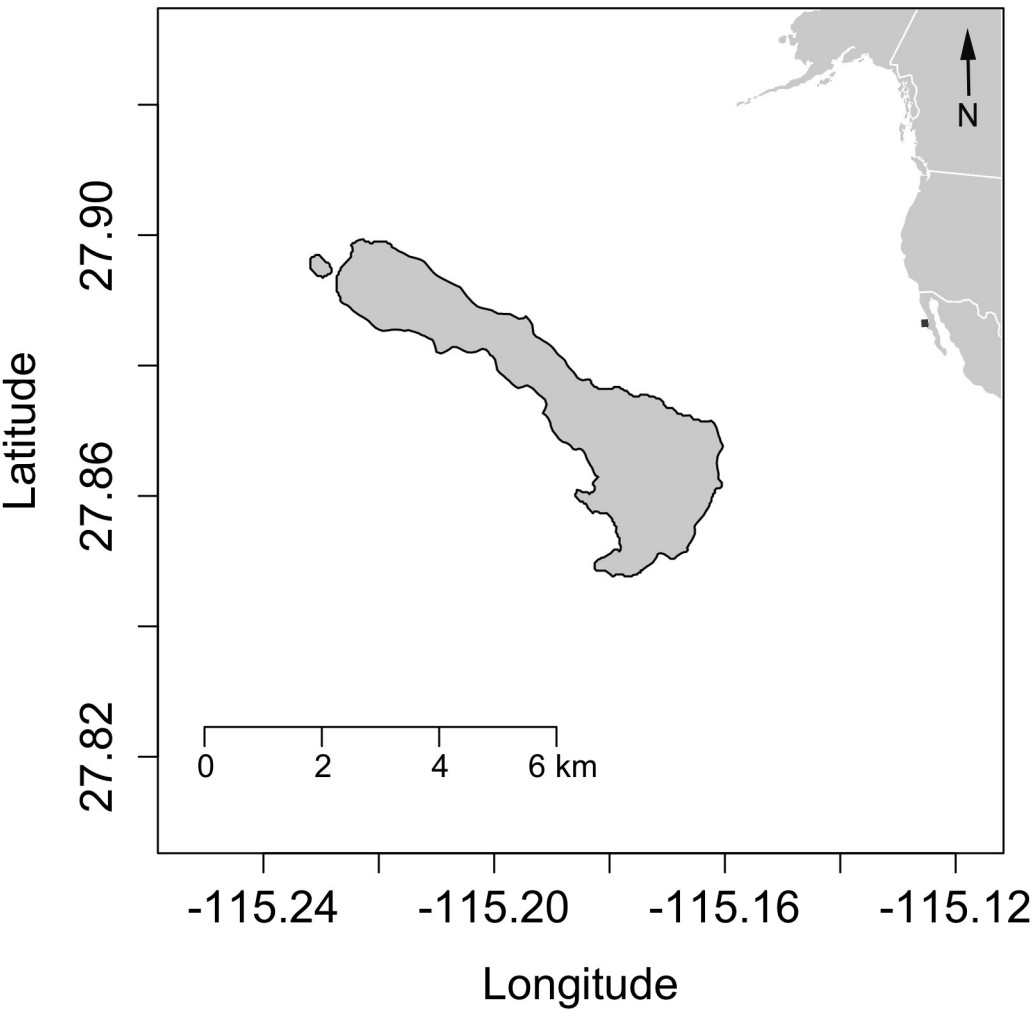

**Fig 1. Map of Isla Natividad off the coast of North America.** The map was generated with r package "raster" License: CC BY-SA 4.0 https://github.com/rspatial/rspatial-raster-web.

### Model parameterization

The incorporation of OA relationships with ecosystem functional groups for this model followed the approach of Busch and McElhany (2016) [34], which synthesized 393 papers reporting the sensitivity of temperate species to changes in seawater carbon chemistry. Their methods for quantifying the sensitivity of functional groups found in the California Current (CC) were based on how well published studies related to functional groups in pH conditions of the CC, experimental design and quality, and the type of variables measured. Quantitative relationships between functional groups and pH sensitivity were provided through 'relative survival scalars', which were derived from qualitative scoring of three factors, 1) direction of pH effect from each study, 2) total amount of evidence available, and 3) level of agreement among studies. These scores were scaled relative to the most sensitive functional group's score to arrive at a relative survival scalar, which provides a linear relationship between survival and pH. Additionally, "high" and "low" pH sensitivity estimates were calculated to provide

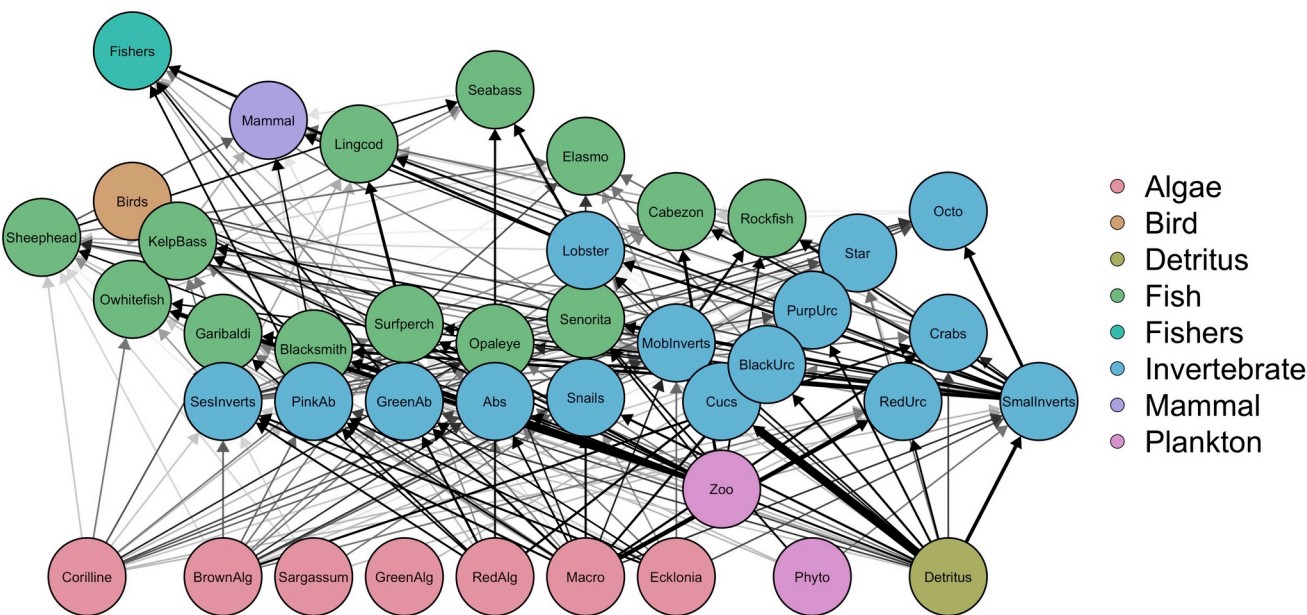

**Fig 2. Food web network diagram of the Isla Natividad model.** Lines represent diet interactions and the strength of each interaction is represented by the line color and thickness (dark > light, thick > thin). Major ecosystem groups are represented by different colors.

uncertainty bounds on potential ecosystem model output. These alternative estimates were used by Busch and McElhany (2016) [34] to calculate upper and lower boundary pH survival scalars.

Due to the focus on CC organisms, the functional group delineations given in Bush and McElhany (2016) closely matched the model component aggregations used in the Isla Natividad EwE model. For example, OA relationships developed for large crustaceans and nearshore sea urchins were applied to the crab/lobster and purple urchin components in the EwE model, respectively. However, in many cases the EwE model included more detail with respect to representing specific trophic guilds or species, and this was accounted for by assigning OA relationships from the closest-fitting broader functional group. For example, the macroalgae OA relationship from Busch and McElhany (2016) [34] was applied to brown algae, red algae, and *Macrocystis pyrifera*. Similarly, the fish OA relationship was applied to lingcod and sea basses, etc. There were also a few instances where the Busch and McElhany (2016) [34] functional groups included more detail than the EwE model components. In these situations, multiple functional group relationships from Busch and McElhany (2016) [34] were averaged together to represent the EwE model component. For example, the average OA relationship from large, meso, and small zooplankton were used to represent zooplankton in the EwE model. A full list of the Busch and McElhany (2016) [34] functional groups used to represent EwE model components can be found in Table 1.

The application of OA relationships derived from Busch and McElhany (2016) [34] to an ecosystem model they were not specifically developed for adds a few sources of uncertainty, which may impact results. For example, the OA relationships developed by Busch and McElhany (2016) [34] were created using species that span the entire CC, and therefore, their application to a Baja California model may not fully represent dynamics associated with warmer waters. Aggregation of species' responses into encompassing functional groups (e.g., fish or benthic herbivorous grazers) by Busch and McElhany (2016) [34] means that specific species and genera included in the EwE model, such as cabezon or purple sea urchin, may not be fully

**Table 1. Species and functional groups used in the EwE Isla Natividad model along with the OA survival scalars associated with their equivalent functional group taken from the Busch and McElhany (2016) meta-analysis.**

| EwE Groups | Scientific Name | Meta-Analysis Groups | Biomass (tons/km²) | Survival Scalar | | |
| --- | --- | --- | --- | --- | --- | --- |
| | | | | Standard | Upper Boundary | Lower Boundary |
| Birds | | | 0.08 | NA | NA | NA |
| Marine Mammals | | | 0.8 | NA | NA | NA |
| Lingcod | *Ophiodon elongatus* | Fish | 0.79 | 0.25 | 0.51 | -0.01 |
| Black Sea Bass | *Stereolepis gigas* | Fish | 20.49 | 0.25 | 0.51 | -0.01 |
| Elamobranchs | *Heterodontus francisci, Rhinobatos productos, Squatina californica* | Small demersal sharks | 89.25 | 0.29 | 0.58 | -0.01 |
| Sheephead | *Semicossyphus pulcher* | Fish | 20.6 | 0.25 | 0.51 | -0.01 |
| Ocean whitefish | *Caulolatilus princess* | Fish | 2.27 | 0.25 | 0.51 | -0.01 |
| Cabezon | *Scorpaenichthys marmoratus* | Fish | 0.02 | 0.25 | 0.51 | -0.01 |
| Rockfishes | *Sebastes* spp. | Fish | 0.14 | 0.25 | 0.51 | -0.01 |
| Basses | *Paralabrax clathratus, Paralabrax nebulifer* | Fish | 4.95 | 0.25 | 0.51 | -0.01 |
| Garibaldi | *Hypsypops rubicundus* | Fish | 18.69 | 0.25 | 0.51 | -0.01 |
| Damselfish | *Chromis punctipinnis* | Fish | 8.2 | 0.25 | 0.51 | -0.01 |
| Surf perch | *Anisotremus davidsonii, Rhacochilus vacca, Embiotoca jacksoni* | Fish | 2.46 | 0.25 | 0.51 | -0.01 |
| Opaleye | *Girella nigricans* | Fish | 14.54 | 0.25 | 0.51 | -0.01 |
| Señoritas | *Halichoeres semicinctus, Oxyjulis californica* | Fish | 3.3 | 0.25 | 0.51 | -0.01 |
| Large crustaceans | *Loxorynchus grandis* | Crabs | 0.5 | 0.7 | 0.95 | -0.01 |
| Sessile Invertebrates | *Crassadoma gigantea*, Anemone spp, *Muricea* spp, *Leptogorgia chilensis* | Meiobenthos shallow benthic filter feeders / soft corals | 7.63 | 0.39 | 0.74 | -0.01 |
| Pink abalone | *Haliotis currugata* | Benthic Herb Grazers | 20.79 | 1 | 1.22 | -0.01 |
| Green abalone | *Haliotis fulgens* | Benthic Herb Grazers | 13 | 1 | 1.22 | -0.01 |
| Other abalones | *Haliotis rufescens, Haliotis sorenseni* | Benthic Herb Grazers | 0.09 | 1 | 1.22 | -0.01 |
| Snail | *Megastraea turbanica, Megastraea undosa* | Benthic Herb Grazers | 476.4 | 1 | 1.22 | -0.01 |
| Mobile invertebrates | *Megathura crenulata, Neobernaya spadicea, Kelletia kelletii* | Meiobenthos / benthic herbivorous grazers / carnivorous infauna | 5.87 | 0.7 | 0.98 | -0.01 |
| Octopuses | *Octopus bimaculatus* | Humbolt Squid /market squid | 0.04 | 0.22 | 0.52 | -0.01 |
| Lobster | *Panulirus interrputus* | Crabs | 103.11 | 0.7 | 0.95 | -0.01 |
| Sea cucumber | *Parastichopus parvimensis* | Deposit Feeders | 10.63 | 0.37 | 0.72 | -0.01 |
| Sea stars | *Patiria miniata, Pisaster giganteus, Pycnopodia heliantoides* | Sea Stars | 0.22 | 0.21 | 0.58 | -0.01 |
| Purple urchin | *Strongylocentrotus purpuratus* | Nearshore sea urchins | 0.69 | 0.2 | 0.6 | -0.01 |
| Black urchin | *Centrostephanus coronatus Arbacia stellata* | Nearshore sea urchins | 1.55 | 0.2 | 0.6 | -0.01 |
| Red urchin | *Mesocentrotus franciscanus* | Nearshore sea urchins | 1.88 | 0.2 | 0.6 | -0.01 |
| Other small invertebrates | *Cancer* spp. | Meiobenthos | 41.05 | 0.07 | 0.38 | -0.01 |
| Coralline algae | | Coralline algae | 30.78 | 0.77 | 1.04 | -0.01 |
| Brown algae | *Cystoseira osmundacea, Laminaria* sp. | Macroalgae | 5.22 | -0.1 | 0.29 | -0.01 |
| Sargassum | *Sargassum horneri, Sargassum muticum* | Macroalgae | 3.34 | -0.1 | 0.29 | -0.01 |
| Green algae | | Macroalgae | 0.2 | -0.1 | 0.29 | -0.01 |
| Red algae | | Macroalgae | 4.28 | -0.1 | 0.29 | -0.01 |
| Giant kelp | *Macrocystis pyrifera* | Macroalgae | 1064.03 | -0.1 | 0.29 | -0.01 |
| Subcanopy kelp | *Ecklonia arborea, Pterygophora californica,* | Macroalgae | 103.89 | -0.1 | 0.29 | -0.01 |
| Zooplankton | | Large zooplankton / meso/ micro | 5.38 | 0.4 | 0.75 | -0.01 |
| Phytoplankton | | Small phytoplankton / large | 2.51 | -0.02 | 0.38 | -0.01 |

For EwE functional groups with more than one functional group equivalent, survival scalars were calculated as the mean value across groups.

**Table 2. Ecological indicators related to the food web structure of the Isla Natividad model, statistics, and network flow parameters.**

| Statistic | Value | Units |
|---|---|---|
| Sum of all consumption | 1576.837 | t/km$^2$/year |
| Sum of all exports | 691.1761 | t/km$^2$/year |
| Sum of all respiratory flows | 1042.608 | t/km$^2$/year |
| Sum of all flows into detritus | 1033.012 | t/km$^2$/year |
| Total system throughput | 4343.633 | t/km$^2$/year |
| Sum of all production | 1891.035 | t/km$^2$/year |
| Mean trophic level of the catch | 1.424436 | |
| Gross efficiency (catch/net p.p.) | 0.007275 | |
| Calculated total net primary production | 1611.629 | t/km$^2$/year |
| Total primary production/total respiration | 1.545766 | |
| Net system production | 569.0203 | t/km$^2$/year |
| Total primary production/total biomass | 4.000377 | |
| Total biomass/total throughput | 0.092749 | t/km$^2$/year |
| Total biomass (excluding detritus) | 402.8692 | t/km2 |
| Total catch | 11.7239 | t/km$^2$/year |
| Connectance Index | 0.155819 | |
| System Omnivory Index | 0.14103 | |
| Shannon diversity index | 2.038866 | |

represented or their responses to OA may be overshadowed by species for which there is more information available in the literature. This should be refined as future research provides better data. There is also the reverse situation where species are aggregated into groups within the Ecopath model (e.g., sessile invertebrates) for which relationships from Busch and McElhany (2016) [34] had to be combined, such as meiobenthos, benthic herviorous grazers, and carnivorous infauna. It has been shown that there can be considerable variation in the response of individual species within larger taxa [57] and this should be taken into consideration when interpreting model results. Furthermore, while Busch and McElhany (2016) [34] described their sensitivity scalars as linear relationships, many species have displayed nonlinear responses to OA [58]. Unfortunately, not enough information exists to accurately quantify and scale those nonlinear relationships across entire functional groups.

When scaling up OA forcing to food web models consisting of biomass components and linked energy flows, impacts can be primarily applied through three major mechanisms; 1) production, 2) direct mortality (a subcomponent of total mortality), and 3) trophic interactions. These forcing mechanisms represent three major mathematical characteristics of network model structure and function that have been shown to strongly influence nonlinear behaviors related to the stability and resilience of a system to perturbations; 1) input, 2) storage, and 3) output [59]. Using OA relationships to affect the production of a model component in the EwE model directly impacts the amount of energy input moving into that particular functional group as well as the entire ecosystem. This can be done in EwE by changing the Production Rate for primary producers or the Search Rate for consumers. Search Rate is a behavioral response representing the volume searched per unit time by a predator, but also serves as a proxy for metabolic changes in production for consumers by modifying the flow of energy from one model component to another [60]. Storage in EwE models is represented by biomass pools and these can be directly forced by OA through the Other Mortality parameter. Other Mortality is a subcomponent of Total Mortality that can be used to represent contributions to

mortality that fall outside natural mortality rates, predation, or fishing mortality. Total Mortality is derived in EwE through a variety of natural growth parameters, predation rates, and fishing rates, while the contribution of Other Mortality can be manually added to this calculated at any timestep. Finally, the energy output of a model component can be forced in EwE through the Vulnerability parameter, which can be used to scale the flow of energy through trophic interactions. Vulnerability is a measure of how susceptible a prey species is to predation and directly impacts the energy output from model components [44]. Generally, forcing model production tends to have the biggest impact on driving model food webs, followed by changing interaction strengths and biomass pools through mortality [61], but these forcing mechanisms have not been individually compared in the context of OA.

The incorporation of OA forcing by the Production and Vulnerability parameters was done through relative scaling to the pH sensitivity curves developed in Busch and McElhany (2016) [34]. In other words, the original values for these parameters were assumed to occur at a pH of 8.0 (normal conditions), which coincides with a pH sensitivity factor value of 1. As pH conditions changed, resulting in an increase or decrease of the pH sensitivity factor based upon the linear slope of the relationship, the baseline values of Production and Vulnerability were multiplied by this factor to simulate changes resulting from OA.

Since Other Mortality is not directly included in the original EwE model, relative scaling of that parameter was not an option. Instead, methods developed in Marshall et al. (2017) were applied to quantify Other Mortality. The relationships developed in Busch and McElhany (2016) [34] were created with the goal of incorporating them into an Atlantis model of the CC, which was published in Marshall et al. (2017). In that study, direct mortality effects were calculated as

$$M^{pH} = (8 - pH)*(-0.1*Sf)$$

where $M^{pH}$ is mortality at a given environmental pH, $Sf$ is the survival scalar developed in Busch and McElhany (2016) [34], and -0.1 is a scaling factor derived in Marshall et al. (2017). After testing a number of scaling factors, they chose -0.1 because it led to an induced mortality rate twice that of the maximum predation mortality on benthic invertebrates when pH dropped from 8.0 to 7.0. For this study, we used a scaling factor of -0.2 because it led to the same proportional increase in Other Mortality with respect to predation mortality as in Marshall et al. (2017) for the equivalent functional group within the EwE model.

All three forcing mechanisms were also applied at the same time to create Combined forcing simulations. Combined forcing best represents real world conditions as OA impacts a variety of ecosystem dynamics simultaneously. This was done by activating Production, Other Mortality, and Vulnerability forcing functions in conjunction with one another during the same model simulation. By modeling each forcing mechanisms individually at first, it is possible to assess their relative importance in contributing to OA impacts, while Combined forcing can elucidate potential feedbacks between forcing mechanisms and better capture realistic ecosystem-level change.

OA interacts with other environmental constraints, such as temperature and oxygen. Currently, individual quantifications of the respective relationships between temperature, oxygen, and all of the functional groups used in this study do not exist. Furthermore, while there is extensive literature investigating the individual effects of temperature, oxygen, and OA on marine organisms, very few studies have addressed their interactive effects [62–65]. As such, temperature and oxygen are not included in this study. External trophic constraints, such as disease, were also not included in this analysis for similar reasons, although climate change is predicted to increase the incidence of disease [66]. Although different species will have

different capacity for evolutionary responses to future OA [67], little is known about this capacity for most species so possible evolutionary responses have not been incorporated into this study.

## Simulations

OA impact relationships were used to simulate a drop in pH from 8.0 to 7.5 over the course of 50 years. This decrease in pH was selected because it encompasses the predicted pH levels of 7.8 in nearshore areas of the CC by 2050 [68] and then continues to decrease in order to identify any potential major ecosystem shifts resulting from extreme stress. Simulations were run using each forcing type (Production, Other Mortality, and Vulnerability) individually as well as all forcing types simultaneously under Combined forcing. Besides responses on individual taxa, ecosystem-level indicators were used to assess a variety of emergent ecosystem properties related to ecological and economic health, including Ecosystem Biomass (summed biomass of all ecosystem components), Commercial Biomass (summed biomass of commercially fished components), Biodiversity (Shannon diversity), and Average Trophic Level of the community (ATL). Kolmogorov-Smirnov tests were also used to test for nonlinearity in the differences between the additive effects of each individual forcing mechanism and the combined simulation. The time series of differences were derived by first calculating the differences between the baseline model values and time series for each forcing mechanism, adding them together, and then calculating the difference between those added time series and the difference between the Combined simulation time series and the baseline model. The above analyses were repeated using the high and low boundary pH sensitivity scalars for each forcing type individually as well as combined in order to establish upper and lower confidence boundaries across model simulation results.

## Results

Simulating a 0.5 drop in pH over 50 years resulted in a diverse set of changes across species and functional groups for all forcing types. Within each simulation, there was a wide distribution of responses, which were directionally consistent across forcing types, but with varying extents of change. Carbonate mineralizing groups such as coralline algae, abalone, snails, and lobsters displayed the largest decreases in abundance while macroalgae and some larger fish species displayed the largest increases. Black urchins also showed an increase in biomass despite being a carbonate mineralizing group. However, there were some distinctions in abundance change for specific species (e.g., brown algae and snails) and functional groups across simulations resulting from differences in trophic interactions driven by each OA forcing type. The rest of this section focuses on a subset of ecologically and commercially important species and functional groups that synthesize ecosystem-scale responses to OA forcing, which include giant kelp, brown algae, coralline algae, pink abalone, black urchins, snails, lobster, cabezon, and the basses.

### Functional group responses

With Production forcing, by the end of the simulation lower trophic level groups, such as giant kelp and brown algae, increased in abundance by approximately 16% and 71%, respectively (Fig 3A) due to slightly increase in production from OA as well as decreased predation from snails. Snails were particularly influential in this ecosystem because they represent the second-highest biomass, after giant kelp. Coralline algae displayed a large decrease in abundance by 62% caused by decreased production from OA as well as increased predation from crabs, which have a high biomass in this model. Snails dropped by approximately 33% as a direct

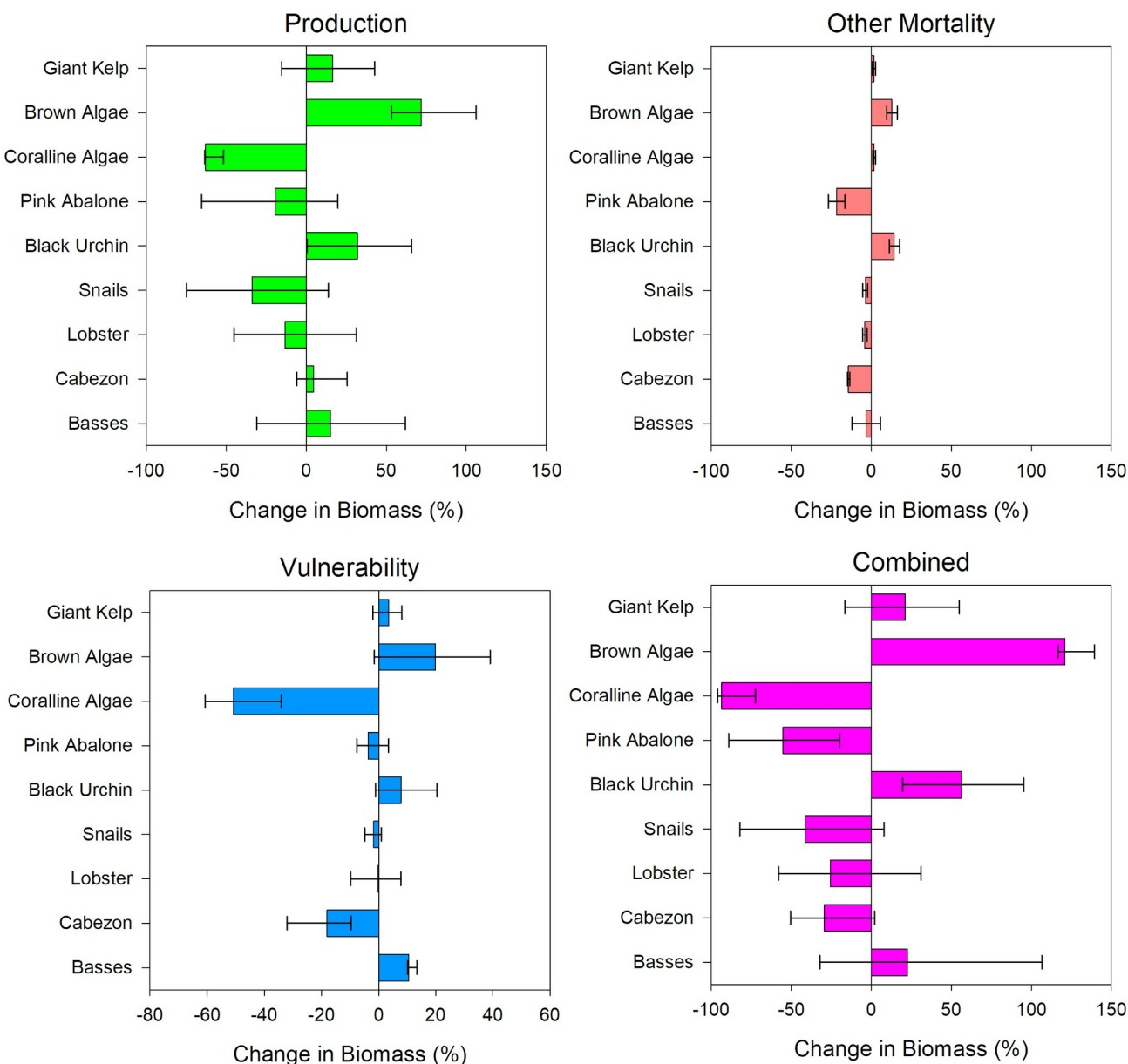

**Fig 3. Plot of percent biomass change across giant kelp, brown algae, coralline algae, pink abalone, snails, lobster, cabezon, and bass species and functional groups under production, other mortality, vulnerability and combined forcing.** Error bars represent percent biomass change under upper and lower boundary survival scalars. Upper boundary scalar simulations are associated with error bars of the same directional change of the normal survival scalar while lower boundary simulations are associated with error bars of the opposite direction change (ex. *Macrocystis* and brown algae upper boundary simulations are represented by the right error bars, while snails and lobster upper boundary simulations are represented by the left error bars).

effect of OA; this decrease had a large trophic impact on other groups in the ecosystem, especially brown algae. Lobster and pink abalone displayed similar decreases in abundance of 13% and 20% from OA, respectively, but increased predation rates on lobster by sheephead also contributed. Black urchin abundance increased by 32% as a result of increases in prey groups such as giant kelp, brown algae, and red algae. Higher trophic level fish groups increased in abundance, with cabezon growing by 4% and basses by 15%.

Applying the upper and lower confidence boundary pH sensitivity scalars to the Production forcing simulations resulted in a wide range of responses across most ecosystem components, with only coralline algae showing little sensitivity to these boundaries, and only coralline algae, brown algae, and black urchin having upper and lower boundaries that did not include zero. For other components, there were significant amplifications of abundance change under the high sensitivity pH boundary and opposite abundance changes under the low pH sensitivity boundary. The large distribution in species and functional group responses were due to the sensitivity of ecosystem models to changes in production rates of primary producers, especially giant kelp, which were subject to significant changes when using the upper or lower boundary pH sensitivity scalars.

Under Other Mortality forcing (Fig 3B), which represents a sub-component of overall mortality (natural + predation + fishing + other), abundance changes differed considerably from the pattern under Production forcing. Giant kelp increased in abundance minimally and brown algae increased by only 13%. Rather than a large decrease, coralline algae abundance increased slightly; the decrease in the predation rate by snails was enough to override the deleterious effects of OA. Pink abalone decreased in abundance by 22% due to the direct effects of Other Mortality (i.e., mortality from OA), while the other shell forming species, such as snails and lobsters, experienced only a small decrease in abundance of 4%. Black urchin abundance increased by 14% due to increased availability of macroalgae. Both higher trophic level fish groups decreased in abundance. Cabezon dropped by approximately 14%, which followed decreasing trends in pink and green abalone prey species abundance, while basses dropped by approximately 3%. Since Other Mortality is just one piece of Total Mortality, its impact can be overshadowed by other components, such as predation mortality, through trophic interactions. Consequently, the direct effect of OA through Other Mortality led to relatively small changes in species abundance.

In contrast to the Production forcing, applying the upper and lower confidence boundary pH sensitivity scalars to the Other Mortality simulations resulted in only very small differences in abundance across components, with the direction of change remained consistent across all simulations except for the basses.

Under Vulnerability forcing (Fig 3C), most responses were similar in direction to Production forcing, but there were distinct differences in the extent of change for brown algae, snails, and black urchin abundances, which were much less. Cabezon abundance also decreased by 18% compared to an increase of 4% under Production forcing.

Applying the upper and lower confidence boundary pH sensitivity scalars to the Vulnerability forcing simulations did not change the directional responses from the base sensitivity simulation except for species and groups that had a <10% change in abundance, such as kelp, pink abalone, black urchin, snails and lobster. However, this was not the case for brown algae, where the low-sensitivity scalar included negative values despite the base result of a 20% increase in abundance.

Combining all forcing types (Fig 3D) generally led to the same pattern of abundance change as Production forcing, but with more extreme responses. Similar to all forcing types, giant kelp and brown algae abundance increased, but to a larger magnitude of 21% and 121%, respectively. This was due to the combined influence of direct OA effects as well as decreased predation rates across multiple functional groups and species. Coralline algae abundance decreased by 93% due to a similar combination of direct OA effects and predation rates. Pink abalone abundance showed a larger decrease in abundance of 55% when compared to any individual forcing type, mainly due to direct OA effects. Snails and lobsters also displayed higher magnitude decreases of 41% and 25%, respectively, resulting from OA, decreased prey availability, and increased predation rates by sheephead. Black urchins increased in abundance by

approximately 56%, mainly due to increased macroalgae prey availability. A combination of decreased prey abundance, direct mortality, and higher lingcod predation led to a decrease in cabezon abundance by 29%, in contrast to a slight increase under Production forcing, while the abundance of basses increased by approximately 23% as a result of decreased predation from elasmobranchs and increased prey abundance.

Similar to Production forcing, applying the upper and lower confidence boundary pH sensitivity scalars resulted in a wide range of responses across many ecosystem components due to the sensitivity of model components to primary production rates at either extreme. However, the two largest responses, brown algae and coralline algae, had relatively small confidence bounds and thus are relatively reliable estimates.

## Comparing forcing types

In general, the directional response of functional groups to OA was consistent across forcing types. However, there were minor shifts in the opposite direction in cabezon abundance under Production forcing as well as basses and coralline algae abundance under Other Mortality forcing. The Production forcing simulations resulted in some of the largest increases in abundances across model components due to the sensitivity of ecosystem structure and function to changes in the production rates of primary producers. OA impacts under Other Mortality forcing, which directly affects biomass storage, resulted in the lowest magnitude of change across model components when compared to other forcing types. This was likely due to the fact that the additional mortality for each taxon resulting from OA was relatively small compared to the effects of predation and production on biomass through the other forcing types over this pH range. Vulnerability forcing resulted in OA impacts similar in magnitude to Other Mortality, excluding coralline algae, which displayed a significantly larger decrease. Combining all forcing types together led to the largest shifts in biomass, showing additive changes across the individual impacts of each forcing type.

## Ecosystem responses

Changes in Ecosystem Biomass across all forcing types displayed varying responses at the beginning of each simulation, but consistently showed very little change over the rest of the model run (Fig 4A). Production forcing, and consequently Combined forcing, had noticeably larger increases in Ecosystem Biomass early in the simulation because of increased production rates for macroalgae. Other Mortality forcing showed almost no change throughout the simulation. The Vulnerability forcing simulation showed small cycles in variation taking place at approximately 10 year intervals due to varying densities of available prey, which caused predators to experience slight shifts in preference that manifested through biomass fluctuations. Other Mortality forcing also displayed minor fluctuations in Ecosystem Biomass at a higher frequency due to the same mechanism.

Commercial Biomass responded differently under each forcing type simulation over the 50 year period (Fig 4B). Similar to Ecosystem Biomass, there was an increase in Commercial Biomass for the Production and Combined forcing types at the beginning of the simulations. The initial observed increases for Production and Combined forcing were mostly due to the elevated biomass of macroalgae, which make up a large portion of commercial fleet harvest. Commercial Biomass then decreased over the remainder of the simulations because of declining snail abundance, dropping by as much as 23 tons/km$^2$. In contrast, under Vulnerability forcing, Commercial Biomass steadily increased by approximately 4 tons/km$^2$, which was mainly due to snail abundance experiencing a slight decline while commercial macroalgae increased. Other Mortality showed a steady decrease in Commercial Biomass over the simulation period,

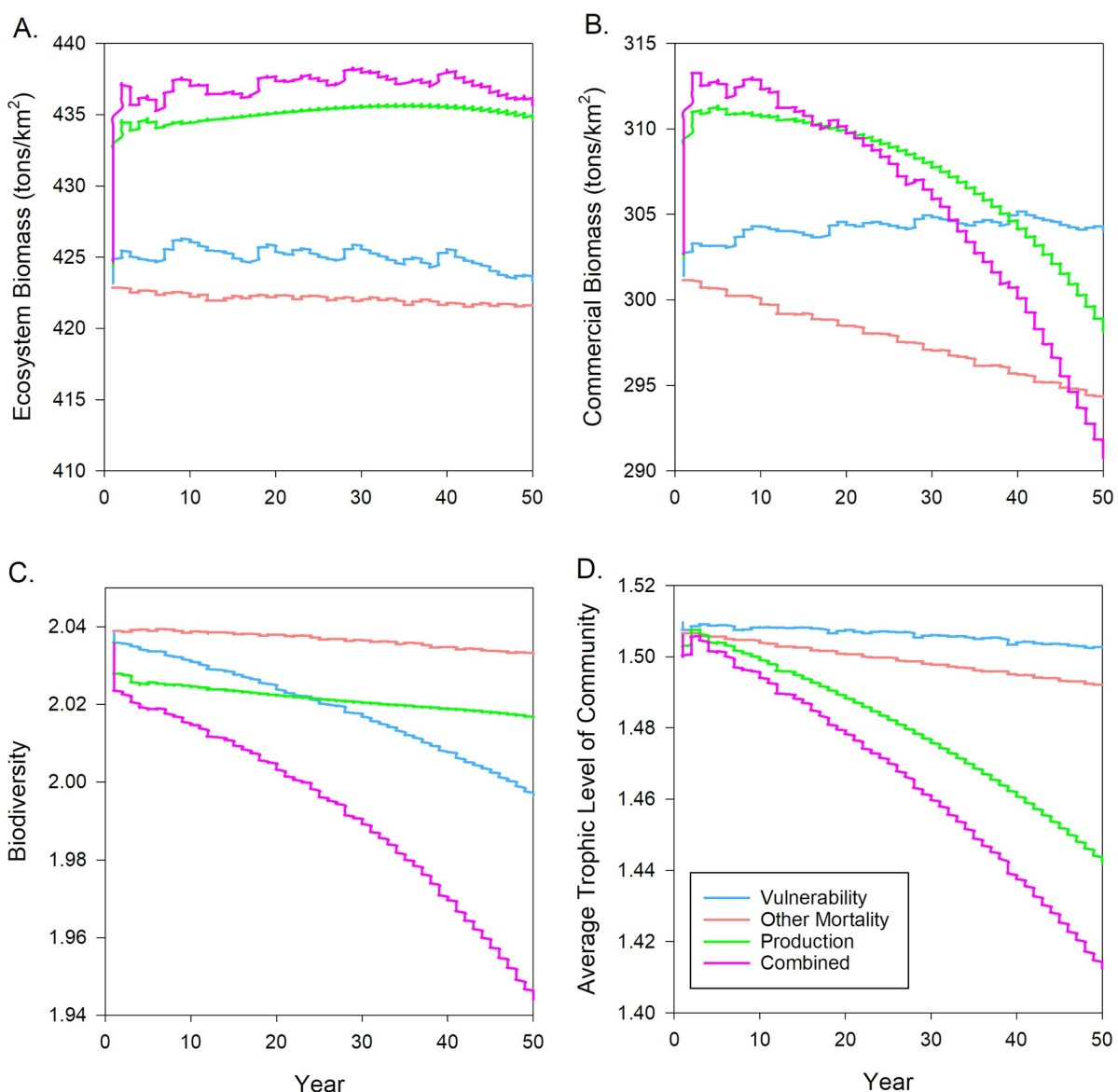

**Fig 4. Plot of changes in total biomass, commercial biomass, biodiversity, and average trophic level of the community ecological indicators across production, other mortality, vulnerability, and combined forcing simulations over 50 years, with pH dropping from 8.0 to 7.5.**

consistent with snail abundance biomass change. The rate of change was linear for Vulnerability and Other Mortality forcing, but displayed an exponential decrease under Production and Combined forcing. There were also small cycles in variation across forcing types similar to Ecosystem Biomass.

Biodiversity generally decreased across all forcing types and was mainly driven by a redistribution of biomass across fewer species and functional groups (Fig 4C). Production forcing displayed a sharp decrease in Biodiversity at the beginning of the simulation due to elevated primary producer biomass, but showed very little change past that over the 50-year period. Other Mortality forcing showed only minor changes throughout the simulation as a result of only small changes to the biomass of model components. Biodiversity under Vulnerability and

**Table 3. Kolmogorov-Smirnov test results for nonlinearity of the differences between the additive effects of production, other mortality, and vulnerability forcing against the combined forcing.**

|                     | K-S Statistic | P-value   |
|---------------------|---------------|-----------|
| Biomass             | 0.4054        | 1.24E-86  |
| Commercial Biomass  | 0.4417        | 9.81E-103 |
| Biodiversity        | 0.4989        | 5.85E-131 |
| Trophic Level       | 0.4993        | 3.45E-131 |

Combined forcing decreased to a larger extent, caused by decline in pink, green, and other abalone functional groups as well as some higher trophic level fish.

With respect to changes in the Average Trophic Level of the Community, all forcing types showed a general decrease over the 50 year period (Fig 4D). The decrease was most noticeable for Production and Combined forcing due to the increase in primary producer biomass resulting from OA. Vulnerability and Other Mortality forcing experienced similar increases in the biomass of primary producers, leading to a drop in the Average Trophic Level, but to a lesser extent.

Overall, the ecosystem-level responses to OA were generally consistent in shape and magnitude across forcing types. However, there were minor differences in the shape and extent of these trends. Production forcing tended to have the largest impact on ecosystem properties due to the sensitivity of ecosystem structure and function to changes in total energy input, expressed as the production rates of primary producers. OA impacts under Other Mortality forcing resulted in the lowest magnitude of change when compared to other forcing types, indicating that the increased direct mortality resulting from OA was not as significant as other forcing impacts with respect to ecosystem responses under the pH range used for these simulations. Vulnerability forcing resulted in more unique patterns across ecosystem properties due to the complex indirect effects resulting from changing trophic interaction strengths. In contrast to the additive effects of combining forcing types together on the abundance of individual species and functional groups, Combined forcing led to nonlinear amplifications or reductions across all ecosystem-level properties (Table 3). This is commonly observed in complex systems when structural changes scale over multiple dimensions (e.g. changes in species/functional group interactions scaled to emergent ecosystem properties).

## Discussion

This study shows how OA has the potential to impact nearshore kelp ecosystems, but the shape and extent of that impact is strongly influenced by the specific physiological interactions of OA with individual species and functional groups. Due to the diverse ways in which OA stress manifests at both individual and population levels, ecosystem-level effects vary and can display nonlinear patterns. Our modeling approach allows us to predict the different impacts of specific OA forcing mechanisms as well as how they interact. Isolating OA impacts on growth (Production), mortality (Other Mortality), and predation interactions (Vulnerability) or combining all three mechanisms together leads to a variety of ecosystem responses, with some species and functional groups increasing in abundance and others decreasing. These changes subsequently lead to shifts in ecosystem structure and function with respect to energy flow and organizational complexity. In reality, OA impacts organisms through a wide variety of effects outside of the three forcing factors used here, which in turn, are emergent properties of many intracellular processes occurring simultaneously [26]. Unfortunately, there are not enough data to individually quantify all those effects and there is a high degree of uncertainty surrounding how those mechanisms collectively interact, especially with respect to California

Current functional groups. However, uncertainty pertaining to individual cellular processes can be reduced by utilizing the higher dimensional characteristics that impact entire organisms or populations as a whole, such as production, biomass storage, and mortality [69]. These emergent properties of the system are easier to measure as well as validate and inherently incorporate all of the underlying processes [70]. These forcing types also serve as proxies for model input, output, and connection strengths, which are also the most important mathematical properties to consider when simulating the dynamics of food webs represented as networks.

While previous studies have used EwE to assess the ecological impacts of OA, none have focused specifically on nearshore coastal kelp ecosystems or quantified the role of different physiological forcing mechanisms. Busch et al. (2013) [45] looked at the effects of OA on the Puget Sound food web by linearly forcing the production rates of calcifying functional groups only. They found similar distributions in the direction and overall extent of change across species and functional group abundance, with calcifying groups generally decreasing and other small invertebrate groups increasing. They also observed both counteractive and amplifying interactions between the direct effects of OA and subsequent indirect effects that manifest through ecological shifts, consistent with the results of this study. Guenette et al. (2014) [36] took a different approach to modeling the effects of OA on a western Scotian Shelf EwE model system by qualitatively categorizing a group's vulnerability to OA and using those designations to force production parameters along with the influence of temperature and oxygen. They also found that changes in primary production led to the largest changes in total biomass and that OA generally had a negative influence on large invertebrates and calcareous species. Cornwall and Eddy (2015) [46] utilized an EwE model of a New Zealand temperate coastal ecosystem to predict the impact of OA in conjunction with fishing and marine protected area policies, but only forced production and consumption parameters of two groups (lobster and abalone). They found that OA decreased the biomass of many groups while indirectly benefiting others. It was also shown that fishing had a larger impact on biomass than OA, but that OA effects were more significant in the absence of fishing.

Although there are only a handful of studies using EwE to asses ecosystem-scale impacts of OA, researchers have also used Atlantis models to address these questions. Using an Atlantis model of the northeast US continental shelf, Fay et al. (2017) [14] modeled the effects of OA by independently changing the mortality and production rates of impacted groups. With mortality forcing on all components, the majority of ecosystem groups decreased in biomass, but the increase in some primary producers led to an increase in biomass of specific higher trophic level fish groups and squid. However, production (growth) forcing on all components led to only small changes in the biomass of ecosystem groups. A possible reason for the stronger impact of Production forcing found in our study is the role of algae-based primary production in a kelp ecosystem versus phytoplankton-based production in an open water ecosystem. OA sensitivity relationships from Busch and McElhany (2016) [34] were more positive for algae species when compared to phytoplankton, and for the Isla Navidad kelp forest model, algae make up the majority of total ecosystem production. Marshall et al. (2017) [35] used the same OA sensitivity relationships derived in Busch and McElhany (2016) [34] to assess the impacts of OA on the California Current through mortality forcing. Their results were similar to those of this study, with the majority of groups showing a decrease in abundance and some primary producers and epibenthos showing an increase. They also identified strong indirect effects resulting from ecological shifts. Olsen et al. (2018) [37] looked at a suite of eight Atlantis models to quantify the roles of OA, marine protected areas, and fishing pressure on marine ecosystems. In general, OA had a larger impact than fishery pressure (contrary to Cornwall and Eddy 2015) [46]. They also observed that OA generally led to decreases in total biomass across

ecosystems, but with individual groups such as demersal/pelagic fish, primary producers, and certain benthos groups increasing in biomass.

The results of this modeling study are also consistent with what has been found through in situ ecological observations. Due to the difficulty of artificially changing the pH chemistry of entire ecosystems, scientists have relied on naturally unique conditions, such as volcanic vents, to serve as proxies for broader oceanic changes. For example, Porzio et al. (2011) [10] described changes in macroalgae assemblages across a pH gradient in the Gulf of Naples. They found that the majority of macroalgae species displayed only a 5% decrease as pH fell to 7.8 while some species exhibited enhanced growth. However, coralline algae showed a dispropor-tionate decrease and were completely absent at a pH of 6.7. Kroeker et al. (2013) [57] similarly compared calcareous and fleshy seaweed communities at a $CO_2$ vent site off the coast of Ischia Island. They found that at low pH levels, competition dynamics amplify the shift towards eco-systems dominated by fleshy seaweed, resulting in potential phase shifts as competitive stabili-zation becomes imbalanced. This study highlights the importance of indirect trophic interactions playing a major role alongside the direct effects of OA. Hall-Spencer et al. (2008) [71] studied the same ecosystem, but also highlighted changes in the benthic and macrofauna communities. They found that shell producing organisms, such as gastropods and barnacles, decreased in abundance and were completely absent as pH approached 7.4. Unfortunately, there is a distinct gap in field observation studies assessing upper trophic level responses to ecosystem change resulting from OA.

A notable advantage of ecosystem modeling is the ability to quantify important indirect effects and shifts in community dynamics across OA forcing mechanisms. While the direct effects of changing production or increasing mortality certainly impact abundances, especially with respect to primary producers, the vast majority of consumer species and functional group changes mirrored trends in either predation mortality rates or prey abundance. Interspecies interactions (e.g. predator, prey, and competition) and energy cycling dynamics of the ecosys-tem create negative and positive feedback loops that can indirectly enhance the effects of OA, and this phenomenon has been frequently observed in both marine and terrestrial ecosystems in response to a variety of perturbations [72]. When combining individual forcing types together, these impacts are magnified further. Although the combined effects of all forcing types on specific species and functional groups leads to additive change of abundances in this model, the emergent effects on ecosystem properties are nonlinear (i.e., exponential). In other words, the complexity of natural systems has the potential to exacerbate the impacts of OA on individual populations, but those effects can subsequently have much larger consequences on measures of ecosystem health and stability. Examples of this can be seen in a variety of other circumstances such as bifurcations in coral reefs due to grazing [73] or seagrass ecosystems due to eutrophication [74], but few studies have shown these processes occurring because of OA impacts.

There are some significant ways that this modeling approach could be improved for future applications. The aggregation of species into EwE functional groups means that the responses of more sensitive species could be overshadowed by the larger group. The majority of keystone species used in the Isla Natividad model are represented individually, which helps to minimize uncertainty attributable to over aggregation. However, a modified version of the model could be created that individually captures more target species known to be significantly impacted by pH. The OA response curves derived from Busch and McElhany (2016) [34] were also developed for species spanning the entire CC, so they may not fully represent physiological adaptations associated with species at the edge of its distribution. Furthermore, the choice of response curves by Busch and McElhany (2016) [34] could be modified to better represent spe-cies and functional groups with more extreme responses to pH or nonlinear behaviors. In

other words, the pH relationships were derived from the availability, consistency, and comprehensiveness of available studies, but quantitative differences in responses were qualitatively categorized. For example, if two groups have the same directional response, equal amount of data, and equal agreement, they would have the same pH response curve, but one could potentially have a much stronger response than the other. While this is indirectly accounted for with groups that have a lot of studies available, one could potentially add the degree of response into the methodology to better capture extreme responses. Similarly, using a linear relationship when there is an exponential response to pH could lead to overestimated impacts at low pH levels and underestimated impacts at higher pH levels. Conversely, a logarithmic response could lead to the opposite. The availability of data needed to properly quantify those relationships is still limited, but this will become less of a problem as more studies are published. The potential effects of uncertainty related to these components were reduced by using the upper and lower pH response curves to provide confidence boundaries on model results.

This study serves as an overview of multiple modeling approaches designed to identify the mechanistic differences between different types of OA forcing and their interactive effects. These differences can be significant on both population and ecosystem scales. But it is encouraging to see that when combining OA mechanisms, model predictions align with what has been previously observed across both modeling and field OA studies, providing evidence that the inclusion of more numerous and complex mechanisms of OA impact do not detract from model realism. Yet, the relative importance of each forcing type in impacting natural communities is still unknown. While there is a broad array of laboratory evidence assessing the metabolic, reproduction, and mortality effects of OA on individuals, there is still a prominent gap in studies extending these findings to populations and ecosystem dynamics. Future research efforts need to focus on not only the relative importance between these OA impacts, but also on how they interact with each other, before results can accurately be used to model ecosystem change. Realistic quantification of these relationships is vital in developing comprehensive modeling about future climate change. Researchers studying these mechanisms should keep modeling applications in mind to more effectively facilitate the flow of information for model parameterization. Additionally, large scale field studies looking at a wide variety of functional group abundance and variability will be necessary to validate these models. The study of OA impacts on ecosystem structure and function is still a novel field, but exploratory studies like this provide an important foundation from which to build off.

Furthermore, similar gaps exist in understanding the roles of temperature and OA. There have been few studies assessing interactions between temperature and $CO_2$ [28]. However, in some red algae, $pCO_2$ and temperature interact antagonistically at sub-saturating light intensities, with $pCO_2$ having a stronger effect, whereas they interact synergistically at saturating light intensities, with temperature having the stronger effect [75]. Similarly, little is known about the roles of temperature and low oxygen pressures on community dynamics as well as how they interact with each other and OA. While only pH was used as a forcing parameter in this model, interactions with temperature rise and deoxygenation are likely to amplify these ecological impacts [76]. For example, pH affects a variety of characteristics such as reproduction, behavior, and growth, which decrease an organism's capacity to compensate for metabolic disturbances and result in a narrowing of thermal and hypoxic tolerance windows [33, 63]. Aside from the direct impact of temperature on organismal physiology, increasing temperature will also exacerbate the effects of low pH by stratifying the water column and further decreasing calcium carbonate availability as well as inhibiting reoxygenation of the water column and further decreasing metabolic capacity [77]. Therefore, the ecological changes observed in this study are expected to be more severe under future climate change scenarios because of the influence of multiple stressors.

Due to the potential consequences of these changes on ocean health and natural resource use in the California Current, proper management and mitigation is critical. Since kelp forests are very nearshore ecosystems, coastal management plays a large role due to the impacts of nutrient runoff on stratification, eutrophication, and deoxygenation [78]. By implementing proper water quality policies, the local effects of climate change can be potentially mitigated [79]. Marine protected areas (MPAs), which have the potential to not only protect ecosystem structure and function within specific spatial areas, but also provide important connections to surrounding ecosystems that help maintain food web functions, biodiversity maintenance, and larval dispersal [80, 81] can help achieve these goals. Although the root cause of declining pH in the California Current is a global issue that requires widespread international cooperation to tackle effectively, local management strategies can help create resilient marine ecosystems that have a better capacity to deal with the negative consequences of climate change. A comprehensive understanding of the mechanistic impacts of OA on ecosystem dynamics through quantitative modeling will play an important role in supporting these management efforts.

## Author Contributions

**Conceptualization:** Adam J. Schlenger, Richard F. Ambrose.

**Data curation:** Rodrigo Beas-Luna.

**Formal analysis:** Adam J. Schlenger.

**Funding acquisition:** Richard F. Ambrose.

**Methodology:** Adam J. Schlenger.

**Project administration:** Richard F. Ambrose.

**Resources:** Richard F. Ambrose.

**Software:** Rodrigo Beas-Luna, Richard F. Ambrose.

**Supervision:** Rodrigo Beas-Luna, Richard F. Ambrose.

**Validation:** Adam J. Schlenger.

**Visualization:** Adam J. Schlenger.

**Writing – original draft:** Adam J. Schlenger.

**Writing – review & editing:** Rodrigo Beas-Luna, Richard F. Ambrose.

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
