## [Decision Letter · Decision Letter 0]

16 Oct 2020

PONE-D-20-20343

Forecasting ocean acidification impacts on kelp forest ecosystems

PLOS ONE

Dear Dr. Schlenger,

Thank you for submitting your manuscript to PLOS ONE. After careful consideration, we feel that it has merit but does not fully meet PLOS ONE’s publication criteria as it currently stands. Therefore, we invite you to submit a revised version of the manuscript that addresses the points raised during the review process.

When preparing the revised manuscript, please pay particular attention to the points raised by Reviewer 1 about the interpretation of your results. The reviewer makes some good suggestions on how you should reconsider your interpretation. Please also consider running the extra analyses that Reviewer 1 suggests - the extra analysis could add depth to your paper and make it more applicable to a broader audience.

We look forward to receiving your revised manuscript.

Kind regards,

Bayden D. Russell

Academic Editor

PLOS ONE

Journal Requirements:

Reviewers' comments:

Reviewer's Responses to Questions

**Comments to the Author**

1. Is the manuscript technically sound, and do the data support the conclusions?

Reviewer #1: Yes

Reviewer #2: Yes

2. Has the statistical analysis been performed appropriately and rigorously? 

Reviewer #1: I Don't Know

Reviewer #2: Yes

3. Have the authors made all data underlying the findings in their manuscript fully available?

Reviewer #1: Yes

Reviewer #2: Yes

4. Is the manuscript presented in an intelligible fashion and written in standard English?

Reviewer #1: No

Reviewer #2: Yes

5. Review Comments to the Author

Reviewer #1: This study presents an interesting analysis of the impact of OA on a kelp forest system based on robust modeling techniques, and the authors wrote it well except some careless grammar errors or typos. While previous studies have used EwE to assess the ecological impacts of OA, few have focused specifically on nearshore coastal kelp ecosystems or quantified the role of different physiological forcing mechanisms. So this study is novel and well address this gap. However, there are some concerns relating to methods and interpretation of the results (shown below) which I suggest the authors pay attention to and revise the paper accordingly.

General comments:

1. It might serve the paper better if there are some analyses of the interactive effect of the three forcing types. For instance, the authors could construct another model which only add production and vulnerability forcing to the model and compare the result with the two individual ones mathematically.

2. I found the authors may need to pay attention to the tense of their writing in Results. Using past tense for describing results is a norm in academic literature.

3. The authors should be more careful in interpreting their results in their writing of Abstract and Results. As I found multiple times that there is overinterpretation (e.g. generally consistent which is in fact not so true). I suggest they would make the paper more interesting to highlight the different response of the ecosystem or populations to different OA forcing mechanisms.

Specific comments:

For Introduction

it should add more citations to support some sentences in the 2nd paragraph.

Page 6, line 12, 'select' should be 'selected'

For Methods

Page 8, lines 5 t 6, missing a right bracket.

Page 9, line 4, 'living from' or 'living on'?

Page 15, line 9, grammatical error in 'Search Rate a behavior ..." missing a verb.

For Results

Pages 17 - 18 (first paragraph), inappropriate interpretation: Looking at Fig. 2, I do not think they are generally consistent, and highlighting the differences is more important. For instance, though coralline algae did display consistent decrease, other groups (snails, lobsters) did not or at least showed high levels of sensitivity.

Page 20, 'Under Vulnerability forcing (Figure 2C), ...': I personally found the Vulnerability forcing result are not so similar to Production forcing one, and thus should be explained in much more detail.

Page 20, ' except for brown algae, where the low-sensitivity scalar includes negative values despite the base result of a 20% increase in abundance. ': This is wrong. Other groups were also sensitive to the scalars. Such as pink abalone, black urchin, lobster.

Page 21, lines 19 - 21 & Page 23, line 20 'not as significant as...': This might be wrong for specific groups like cabezon and pink abalone as shown by the figure. The explanation does not tell anything as it is just another way of narrating the result. I worry if the less prominent effect from other mortality forcing was driven by the potential poor choice of parameters to simulate this forcing in the model. At least, based on the methods by Busch and McElhany (2016), the mortality was linearly correlated with ph drop, which as the authors mentioned is not commonly found in nature. In addition, whether this empirical model could be used here in this study was not fully explained by authors. Meanwhile, as the authors also mentioned, shifts in temperature and dissolved oxygen could also interact with the pH but were not considered in this study. So the authors should at least acknowledge these potential contributing factors and interpret their results with serious caution and do not overclaim that direct mortality caused by OA is not significant as compared with other forcing types.

Page 21, line 24: Confusing on 'linearly consistent'. Meanwhile, though it might be consistent with some groups, but definitely not for others (e.g. basses) that had different responses to different forcing factors.

Page 23, line 15: Again, I disagree that they are generally consistent. It is more important and interesting to highlight the difference, which was shown by the figure.

Page 24, line 1 'nonlinear amplifications': This would be better supported with a specific analysis rather than a superficial statement.

For Table 2, the authors misplaced the unit in a few places. For instance, Gross efficiency (catch/net p.p.) should be unitless, while Calculated total net primary production should have a unit.

Reviewer #2: Review for: “Forecasting ocean acidification impacts on kelp forest ecosystems” by Schlenger et al. Overall, the paper was well written and understandable.

Using ecological models to integrate results from different methods. Thus leading to an assessment of Ocean Acidification on kelp community structure. Better understanding of food web dynamics responding to external forcing parameters. Present a quantitative modeling approach to asses the cumulative effects of multiple OA impacts on ecosystem dynamics. Results reveal that impacts from OA is strongly influenced by specific physiological interactions of OC with specific functional groups and species.

Overall comments

Overall, the manuscript is well written and the research is interesting and validates previous similar studies.

I recommend to the authors to read and include the Minireview by Harley et al. 2012 “Effects of climate change on global seaweed communities” – it has some very relevant insight.

To bottom of page 27- the authors could add more detailed data from Kubler et al. 2015 PLoS ONE “Predicting effects of ocean acidification and warming on algae lacking carbon concentrating mechanisms.”

Specific comments

A map of Isla Navidad would be useful to add to the intro – under “ecopath Model”

Figure 1 – the text is very blurry – please make the font bigger and more legible. Figure 2 is also blurry – please make more legible.

Table 1 . “Sea palm” usually refers to Postelsia palameformis. Curious on why do you separate sea palm from brown algae and Sargassum categoryTechnically, Sargassum, Macrocystis, Ecklonia are all brown algae too. Throughout the text and in your figures, you predominantly refer to brown algae – so it is not clear what groups you include. Perhaps there is a better way to make the distinction?

Pg 18 – missing % after 15.

Figure 2 – why only “other mortality” where is overall mortality (natural + predation +fishing +other)? Is it included in Vulnerability?

6. PLOS authors have the option to publish the peer review history of their article (what does this mean?). If published, this will include your full peer review and any attached files.

Reviewer #1: **Yes: **Xiong Zhang

Reviewer #2: **Yes: **Simona Augyte

---

## [Author Response · Author response to Decision Letter 0]

30 Nov 2020

Please read the 'Schlenger et al. Response to Reviewers' for a more clear format and differentiation between responses and reviewer comments.

Reviewer #1: This study presents an interesting analysis of the impact of OA on a kelp forest system based on robust modeling techniques, and the authors wrote it well except some careless grammar errors or typos. While previous studies have used EwE to assess the ecological impacts of OA, few have focused specifically on nearshore coastal kelp ecosystems or quantified the role of different physiological forcing mechanisms. So this study is novel and well address this gap. However, there are some concerns relating to methods and interpretation of the results (shown below) which I suggest the authors pay attention to and revise the paper accordingly.

General comments:

1. It might serve the paper better if there are some analyses of the interactive effect of the three forcing types. For instance, the authors could construct another model which only add production and vulnerability forcing to the model and compare the result with the two individual ones mathematically.

Although we agree with the reviewer that an exploration of the interactive effects between each forcing type would be a useful exercise to better understand the model behavior, we feel that the interpretation of those results would not be ecologically relevant. In a natural setting, there would not be an instance where only two of these mechanisms would be acting in conjunction with each other at the ecosystem-level. Furthermore, the novel utility of this study lies in taking these individual forcing mechanisms, previously looked at independently in other publications, and understanding their combined dynamics in an effort to step as close to ecological realism as possible. We feel that only looking at combinations of two mechanisms at a time would not provide useful insights for those trying to replicate this method in other models or those attempting to validate these findings in the field.

2. I found the authors may need to pay attention to the tense of their writing in Results. Using past tense for describing results is a norm in academic literature.

The verb tense has been changed to past tense as appropriate.

3. The authors should be more careful in interpreting their results in their writing of Abstract and Results. As I found multiple times that there is overinterpretation (e.g. generally consistent which is in fact not so true). I suggest they would make the paper more interesting to highlight the different response of the ecosystem or populations to different OA forcing mechanisms.

We have added a number of revisions clarifying the interpretation of results or adding context highlighting uncertainties in those interpretations. A more detailed breakdown of those changes can be found below.

We have also described a wide variety of changes in abundance across groups and species as well as the ecosystem properties for each individual forcing mechanism in the text. We request a further explanation on where / how the reviewer would like this expanded.

Specific comments:

For Introduction

it should add more citations to support some sentences in the 2nd paragraph.

We have included additional citations to support the role and characteristics of paleo records, experiments, and long term monitoring ecological studies to understand impacts of climate change in marine communities. 

Page 6, line 12, 'select' should be 'selected'

“Select” is used here to indicate a specific group of ecosystems not as a verb.

For Methods

Page 8, lines 5 t 6, missing a right bracket.

Right parenthesis added.

Page 9, line 4, 'living from' or 'living on'?

Yes, this was awkward wording; changed to “subsisting on.”

Page 15, line 9, grammatical error in 'Search Rate a behavior ..." missing a verb.

Added “is.”

For Results

Pages 17 - 18 (first paragraph), inappropriate interpretation: Looking at Fig. 2, I do not think they are generally consistent, and highlighting the differences is more important. For instance, though coralline algae did display consistent decrease, other groups (snails, lobsters) did not or at least showed high levels of sensitivity.

The reviewer is correct in that ‘generally consistent’ does not accurately characterize the patterns observed. This has been changed to ‘directionally consistent’ with a reference to varying extents of change to highlight this difference. While this particular paragraph was designated to provide a quick overview of the results, it was not intended to get into the specific differences between forcing mechanisms, and a more detailed analysis of these differences can be found later on in the results section under ‘Comparing Forcing Types’. However, we did highlight potential differences between simulations in sentence 5 of the first paragraph and added in example species to further address the reviewer’s comment

Page 20, 'Under Vulnerability forcing (Figure 2C), ...': I personally found the Vulnerability forcing result are not so similar to Production forcing one, and thus should be explained in much more detail.

Changed ‘similar’ to ‘similar in direction’. The rest of the paragraph was edited to further highlight those specifics differences in more detail

Page 20, ' except for brown algae, where the low-sensitivity scalar includes negative values despite the base result of a 20% increase in abundance. ': This is wrong. Other groups were also sensitive to the scalars. Such as pink abalone, black urchin, lobster.

Made a number of revisions in the paragraph to more accurately reflect the directional changes in abundances and specifically highlighted the groups the reviewer pointed out.

Page 21, lines 19 - 21 & Page 23, line 20 'not as significant as...': This might be wrong for specific groups like cabezon and pink abalone as shown by the figure. The explanation does not tell anything as it is just another way of narrating the result. I worry if the less prominent effect from other mortality forcing was driven by the potential poor choice of parameters to simulate this forcing in the model. At least, based on the methods by Busch and McElhany (2016), the mortality was linearly correlated with ph drop, which as the authors mentioned is not commonly found in nature. In addition, whether this empirical model could be used here in this study was not fully explained by authors. Meanwhile, as the authors also mentioned, shifts in temperature and dissolved oxygen could also interact with the pH but were not considered in this study. So the authors should at least acknowledge these potential contributing factors and interpret their results with serious caution and do not overclaim that direct mortality caused by OA is not significant as compared with other forcing types.

The explanation on Page 23 was changed to ‘This was likely due to the fact that the additional mortality for each taxon resulting from OA was relatively small compared to the effects of predation and production on biomass through the other forcing types over this pH range.’

The reviewer is correct in his recommendation of caution when interpreting results based upon the mortality parameters presented by Busch and McElhany (2016). The parameters and relationships derived in Busch and McElhany (2016) were specifically developed for the use in ecosystem modeling studies, albeit via ATLANTIS, and therefore were the most up-to-date and applicable to our research. With respect to incorporating those mortality parameters into our model, we followed the methods of Marshall et al. (2017), which used the scalars from Busch and McElhany (2016) in an empirical equation solving for mortality of invertebrates that resulted in rates consistent with the literature. We actually used a higher scaling factor than Marshall et al. (2017) to achieve the same proportional increase in mortality resulting from pH. These methods are described on page 16.

The alternative was to recreate our own literature review across all species and groups, but we still would have been faced with the question of how to handle nonlinear responses due to a lack of overall information in the literature quantifying mortality rates at varying pH levels. As such, we felt that Busch and McElhany (2016) handled that gap in information responsibly. There is a similar gap with respect to the interacting effects of temperature and dissolved oxygen and we address these unknowns and their potential consequences in the last paragraph on page 29. However, it is unknown whether these interactions would also increase the impacts of OA through the Production and Vulnerability mechanisms along with Mortality. As such, we cannot claim that mortality would have a comparatively more significant effect. The same principle applies to the consequences of using linear scaling relationships.

Furthermore, from our experience using a variety of ecosystem models, small changes in the production of low trophic level groups have disproportionately large impacts on the biomass of groups in the ecosystem when compared to increases in top-down mortality of upper trophic species, such as through increased fishing pressure. Therefore, the results of this study are consistent with other modeling research we’ve seen in the past.

Page 21, line 24: Confusing on 'linearly consistent'. Meanwhile, though it might be consistent with some groups, but definitely not for others (e.g. basses) that had different responses to different forcing factors.

‘Linearly consistent’ was changed to ‘showing additive changes across the individual impacts of each forcing type’, which is a more accurate description and takes into account the directional differences displayed in some groups across simulations.

Page 23, line 15: Again, I disagree that they are generally consistent. It is more important and interesting to highlight the difference, which was shown by the figure.

Revised the sentence to ‘generally consistent in shape and magnitude across forcing types’. We believe this to be the case because the overall directional trends and shapes of the relationships do not drastically differ between forcing types. When taking into account the annotated scale on the y-axis, we also believe that the magnitude of each trend does not drastically differ. 

But with that being said, the rest of the paragraph delves into the differences between forcing types in general, and the 4 paragraphs before this breakdown those differences at a more detailed level with respect to each ecosystem property.

Page 24, line 1 'nonlinear amplifications': This would be better supported with a specific analysis rather than a superficial statement.

Per the reviewer’s suggestions, we conducted a series of nonlinearity tests (Kolmogorov-Smirnov) where we 1) calculated the differences between the baseline model and time series for each forcing mechanism, 2) added those differences together, 3) calculated the difference between those added time series and the difference between the Combined simulation time series and the baseline model, 4) then conducted a Kolmogorov-Smirnov test on the linearity of those final time series.

 K-S Statistic P-value

Biomass 0.4054 1.24E-86 

Commercial Biomass 0.4417 9.81E-103 

Biodiversity 0.4989 5.85E-131 

Trophic Level 0.4993 3.45E-131 

Results show that there are distinct nonlinear shifts between how the Combined mechanisms shift ecosystem properties versus the additive shifts from each individual forcing mechanism.

The above table was added into the manuscript and the following was added into the text 

Methods: ‘Kolmogorov-Smirnov tests were also used to test for nonlinearity in the differences between the additive effects of each individual forcing mechanism and the combined simulation. The time series of differences were derived by first calculating the differences between the baseline model values and time series for each forcing mechanism, adding them together, and then calculating the difference between those added time series and the difference between the Combined simulation time series and the baseline model.’

An additional table was added into the results at the end of the ‘Ecosystem Responses’ section presenting these results.

For Table 2, the authors misplaced the unit in a few places. For instance, Gross efficiency (catch/net p.p.) should be unitless, while Calculated total net primary production should have a unit.

Units corrected for Gross Efficiency and Calculated Total Net Primary Production

Reviewer #2: Review for: “Forecasting ocean acidification impacts on kelp forest ecosystems” by Schlenger et al. Overall, the paper was well written and understandable.

Using ecological models to integrate results from different methods. Thus leading to an assessment of Ocean Acidification on kelp community structure. Better understanding of food web dynamics responding to external forcing parameters. Present a quantitative modeling approach to asses the cumulative effects of multiple OA impacts on ecosystem dynamics. Results reveal that impacts from OA is strongly influenced by specific physiological interactions of OC with specific functional groups and species.

Overall comments

Overall, the manuscript is well written and the research is interesting and validates previous similar studies.

I recommend to the authors to read and include the Minireview by Harley et al. 2012 “Effects of climate change on global seaweed communities” – it has some very relevant insight.

We thank the reviewer for highlighting this paper. We have incorporated some of its ideas into parts of the Introduction and Discussion.

To bottom of page 27- the authors could add more detailed data from Kubler et al. 2015 PLoS ONE “Predicting effects of ocean acidification and warming on algae lacking carbon concentrating mechanisms.”

Although we agree that Kubler and Dudgeon (2015) is an interesting and relevant paper, we chose to discuss their results in the paragraph about interactions with temperature rather than the paragraph suggested by the reviewer since it seemed more appropriate there.

Specific comments

A map of Isla Navidad would be useful to add to the intro – under “ecopath Model”

We have included a map of the island indicating the location in relation to North America. 

Figure 1 – the text is very blurry – please make the font bigger and more legible. Figure 2 is also blurry – please make more legible.

Figure 1 (now Figure 2) has been recreated to be more clear

Table 1 . “Sea palm” usually refers to Postelsia palameformis. Curious on why do you separate sea palm from brown algae and Sargassum categoryTechnically, Sargassum, Macrocystis, Ecklonia are all brown algae too. Throughout the text and in your figures, you predominantly refer to brown algae – so it is not clear what groups you include. Perhaps there is a better way to make the distinction?

To avoid further confusion with this group, we have changed the name from “Sea palm” to “Sub canopy kelp”. We initially used this term as Ecklonia arborea is known as “The Southern Sea Palm”. Additionally in table 1, we have listed the genus and species for the most representative groups for all functional groups in the model. 

Pg 18 – missing % after 15.

Added “%.”

Figure 2 – why only “other mortality” where is overall mortality (natural + predation +fishing +other)? Is it included in Vulnerability?

Total Mortality (described on page 15) is a combination of natural mortality, predation mortality, fishing mortality, and other mortality, which are all calculated in the EwE model. Other mortality is a subcomponent of this calculation and is handled separately. In order to further emphasize this point, we added in the following test to page 15.

‘Total Mortality is derived in EwE through a variety of natural growth parameters, predation rates, and fishing rates, while the contribution of Other Mortality can be manually added to this calculated at any timestep.’

---

## [Decision Letter · Decision Letter 1]

26 Jan 2021

PONE-D-20-20343R1

Forecasting ocean acidification impacts on kelp forest ecosystems

PLOS ONE

Dear Dr. Schlenger,

Thank you for submitting your manuscript to PLOS ONE. After careful consideration, we feel that it has merit but does not fully meet PLOS ONE’s publication criteria as it currently stands. Therefore, we invite you to submit a revised version of the manuscript that addresses the points raised during the review process.

We look forward to receiving your revised manuscript.

Kind regards,

Christopher Edward Cornwall, Ph.D.

Academic Editor

PLOS ONE

Additional Editor Comments (if provided):

I have taken over the role of editor for this manuscript. Only one of the two reviewers from the original round accepted the request to review again. This one reviewer indicated they are happy with the revisions. However, I have identified some minor points that need to be addressed by the authors before this manuscript could be accepted for publication. Apologies for the delay, if I had been handling this manuscript from the beginning I would have noted these points initially.

Line 76 – please remove contractions here and anywhere else they occur.

Line 114 – you will need to add in “non-calcareous”, as the impacts of OA on coralline algae, Halimeda, etc are certainly not positive.

Line 178 – Reference 46 is a kelp forest ecosystem

Line 233 – This needs to reference the numbered references in the references section please. Please check elsewhere also for these instances.

Line 547 – Cornwall and Eddy.

Line 570 – This sentence does not make sense entirely, because in Cornwall and Eddy there was only fishing presence/absence (i.e. MPA or not) and OA presence/absence. Do the authors mean OA had a larger effect in the absence of marine protection?

Line 579 – pH 6.7 likely does not represent a scenario we will ever observe in most marine habitats due to OA.

Line 628 – Remove the words” results showed that”.

Line 654 – I consider that any chemical effect of temperature will be very minimal compared to the impact of temperature on organism physiology.

Figures: Figure 3 and 4 appear to be straight form excel, it would be ideal if the authors could improve these.

References: Some of these require formatting, for example, if title names are listed in all capital letters, this needs to be addressed. Please go through each reference to check formatting.

Reviewers' comments:

Reviewer's Responses to Questions

**Comments to the Author**

1. If the authors have adequately addressed your comments raised in a previous round of review and you feel that this manuscript is now acceptable for publication, you may indicate that here to bypass the “Comments to the Author” section, enter your conflict of interest statement in the “Confidential to Editor” section, and submit your "Accept" recommendation.

Reviewer #1: All comments have been addressed

2. Is the manuscript technically sound, and do the data support the conclusions?

Reviewer #1: Yes

3. Has the statistical analysis been performed appropriately and rigorously? 

Reviewer #1: Yes

4. Have the authors made all data underlying the findings in their manuscript fully available?

Reviewer #1: Yes

5. Is the manuscript presented in an intelligible fashion and written in standard English?

Reviewer #1: Yes

6. Review Comments to the Author

Reviewer #1: The manuscript has been well revised and it is now much clearer. I am satisfied with all the responses the author provided.

7. PLOS authors have the option to publish the peer review history of their article (what does this mean?). If published, this will include your full peer review and any attached files.

Reviewer #1: No

---

## [Author Response · Author response to Decision Letter 1]

24 Mar 2021

Response to reviewer comments can be found in the attached document and are easier to read there, but I am copying those responses here as well.

Line 76 – please remove contractions here and anywhere else they occur.

Corrected this and a handful of other instances of contractions throughout the paper

Line 114 – you will need to add in “non-calcareous”, as the impacts of OA on coralline algae, Halimeda, etc are certainly not positive.

Added

Line 178 – Reference 46 is a kelp forest ecosystem

Included

Line 233 – This needs to reference the numbered references in the references section please. Please check elsewhere also for these instances.

Numbers were included alongside all references

Line 547 – Cornwall and Eddy.

Fixed

Line 570 – This sentence does not make sense entirely, because in Cornwall and Eddy there was only fishing presence/absence (i.e. MPA or not) and OA presence/absence. Do the authors mean OA had a larger effect in the absence of marine protection?

What we meant to say here is that the effects of OA were larger on groups when compared to the effects of fishing pressure in Olsen et al whereas the effects of fishing were larger than OA in Cornwall and Eddy. Therefore we removed ‘and marine protected areas’ from the sentence to just focus on OA and fishing pressure. The sentence now reads as 

“In general, OA had a larger impact than fishery pressure (contrary to Cornwall and Eddy 2015).”

Line 579 – pH 6.7 likely does not represent a scenario we will ever observe in most marine habitats due to OA.

That is true and it is why our study did not include such an extreme pH. However, we’re reporting the results of Porzio et al. (2011) to provide evidence of ecological changes along an extended pH gradient. 

Line 628 – Remove the words” results showed that”.

Removed

Line 654 – I consider that any chemical effect of temperature will be very minimal compared to the impact of temperature on organism physiology.

The sentence has been updated to read as

“Aside from the direct impact of temperature on organismal physiology, increasing temperature will also exacerbate the effects of low pH by stratifying the water column and further decreasing calcium carbonate availability as well as inhibiting reoxygenation of the water column and further decreasing metabolic capacity [77]”

Figures: Figure 3 and 4 appear to be straight form excel, it would be ideal if the authors could improve these.

The figures have been reproduced using Sigma Plot. Please let us know if you would like to see any further changes.

References: Some of these require formatting, for example, if title names are listed in all capital letters, this needs to be addressed. Please go through each reference to check formatting.

Fixed

---

## [Editor Report · Decision Letter 2]

29 Mar 2021

Forecasting ocean acidification impacts on kelp forest ecosystems

PONE-D-20-20343R2

Dear Dr. Schlenger,

We’re pleased to inform you that your manuscript has been judged scientifically suitable for publication and will be formally accepted for publication once it meets all outstanding technical requirements.

Kind regards,

Christopher Edward Cornwall, Ph.D.

Academic Editor

PLOS ONE

Additional Editor Comments (optional):

Thank you for your patience, I consider that the manuscript is much improved now that the few pieces of unclear sentences were dealt with.
---

## [Editor Report · Acceptance letter]

12 Apr 2021

PONE-D-20-20343R2 

Forecasting ocean acidification impacts on kelp forest ecosystems 

Dear Dr. Schlenger:

I'm pleased to inform you that your manuscript has been deemed suitable for publication in PLOS ONE. Congratulations! Your manuscript is now with our production department. 

Kind regards, 

on behalf of

Dr. Christopher Edward Cornwall 

Academic Editor

PLOS ONE